# Low-Carbohydrate High-Fat Diet and Exercise: Effect of a 10-Week Intervention on Body Composition and CVD Risk Factors in Overweight and Obese Women—A Randomized Controlled Trial

**DOI:** 10.3390/nu13010110

**Published:** 2020-12-30

**Authors:** Thorhildur Ditta Valsdottir, Bente Øvrebø, Thea Martine Falck, Sigbjørn Litleskare, Egil Ivar Johansen, Christine Henriksen, Jørgen Jensen

**Affiliations:** 1Department of Medicine, Atlantis Medical University College, 0560 Oslo, Norway; 2Institute of Physical Performance, Norwegian School of Sport Sciences, 0863 Oslo, Norway; e.i.johansen@nih.no (E.I.J.); jorgenj@nih.no (J.J.); 3Faculty of Health and Sport Sciences, University of Agder, 4630 Kristiansand, Norway; bente.ovrebo@uia.no; 4Department of Nutrition, Exercise and Sports, University of Copenhagen, Frederiksberg C, 1870 Copenhagen, Denmark; tmkfalck@gmail.com; 5Department of Sports and Physical Education, Inland Norway University of Applied Sciences, 2411 Elverum, Norway; sigbjorn.litleskare@inn.no; 6Institute of Basic Medical Sciences, Department of Nutrition, Faculty of Medicine University of Oslo, 0372 Oslo, Norway; christine.henriksen@medisin.uio.no

**Keywords:** low-carbohydrate high-fat-diet, ketogenic, exercise, fat mass, lean body mass, cardiovascular risk, cardiorespiratory fitness, lipid profile, dietary intervention

## Abstract

We assessed the effect of weight-loss induced with a low-carbohydrate-high-fat diet with and without exercise, on body-composition, cardiorespiratory fitness and cardiovascular risk factors. A total of 57 overweight and obese women (age 40 ± 3.5 years, body mass index 31.1 ± 2.6 kg∙m^−2^) completed a 10-week intervention using a low-carbohydrate-high-fat diet, with or without interval exercise. An equal deficit of 700 kcal∙day^−1^ was prescribed, restricting diet only, or moderately restricting diet and adding exercise, producing four groups; normal diet (NORM); low-carbohydrate-high-fat diet (LCHF); normal diet and exercise (NORM-EX); and low-carbohydrate-high-fat diet and exercise (LCHF-EX). Linear Mixed Models were used to assess between-group differences. The intervention resulted in an average 6.7 ± 2.5% weight-loss (*p* < 0.001). Post-intervention % fat was lower in NORM-EX than NORM (40.0 ± 4.2 vs. 43.5 ± 3.5%, *p* = 0.024). NORM-EX reached lower values in total cholesterol than NORM (3.9 ± 0.6 vs. 4.7 ± 0.7 mmol/L, *p* = 0.003), and LCHF-EX (3.9 ± 0.6 vs. 4.9 ± 1.1 mmol/L, *p* = 0.004). Post intervention triglycerides levels were lower in NORM-EX than NORM (0.87 ± 0.21 vs. 1.11 ± 0.34 mmol/L, *p* = 0.030). The low-carbohydrate-high-fat diet had no superior effect on body composition, $\dot{V}$O_2peak_ or cardiovascular risk factors compared to a normal diet, with or without exercise. In conclusion, the intervention decreased fat mass, but exercise improved body composition and caused the most favorable changes in total cholesterol and triglycerides in the NORM-EX. Exercise increased cardiorespiratory fitness, regardless of diet.

## 1. Introduction

The prevalence of overweight and obesity is high due to chronic positive energy balance. In 2016, over 1.9 billion people were overweight [1]. The condition is characterized by an unfavorable body composition with excess adipose tissue [2] and relatively low lean-body mass [3]. Overweight and obesity are independent risk factors for cardiovascular disease (CVD), and have a high correlation with dyslipidemia [4]. Further, body composition and fat distribution, mainly visceral fat and waist-to-hip ratio (WHR), are correlated with increased risk for CVD [5]. Although the mortality of coronary heart disease has declined during the last four decades, there is evidence for stagnation, especially in women [6]. This increase can to an extent be explained by an increase in weight [7]. Further on, it has been reported that women receive less care in the form of investigation and secondary prevention treatments for CVD than men do [8], and as a result women might have increased risk of death from CVD incidences.

Dietary management with calorie restriction is the most applicable tool for weight-loss [9]. Substantial weight loss (≈10%) with a hypocaloric diet has been shown to reverse many factors correlated with metabolic syndrome [10], and even modest weight-loss (≈5%) improves CVD risk factors [11,12,13]. Low-carbohydrate high-fat (LCHF) diets are popular but previous LCHF studies have often used ad libitum approaches when comparing the diet with other hypocaloric diets, where differences in energy intake and therefore weight-loss, can limit the understanding of the real effect of the different approaches [14]. Meta-analyses of the effect of LCHF diets on CVD risk factors have shown that LCHF diets are superior to control diets with regard to improvements in HDL, VLDL and TG after 6 months–2 years [15,16], and at the same time LCHF diets have also shown a less favorable effect on LDL [15,17]. It has therefore been of a great concern that LCHF diets increase the risk for CVD [15], and it has been questioned if the diet is safe to recommend. However, LCHF diets that induce ketosis have been shown to spare muscle mass [18] and studies on the effect of LCHF diets on body composition have suggested that even with a large weight-loss, lean body mass (LBM) is preserved and fat mass is markedly reduced [19,20].

Exercise can be an efficient tool for weight-loss, although the exercise volumes needed for a large weight loss may not be practical or sustainable in overweight and obese individuals [21]. Centers for Disease Control and Prevention [22] and the Norwegian Health Authorities [23], recommend a dietary energy deficit of approximately 500–1100 kcal∙day^−1^ to achieve a gradual and steady loss of 0.5–1 kg a week. Exercise without weight-loss has a profound effect on physical health in overweight and obese individuals [24,25], and cardiorespiratory fitness (CRF) has a central role in the prevention of CVD, as a greater CRF is associated with lower risk of CVD regardless of body mass index (BMI) [26]. Weight-loss studies that have included exercise have indicated that this method is beneficial for preserving lean-body mass (LBM) [27,28,29]. Maintaining LBM is important to sustain resting energy expenditure [30,31], to retain strength and function, and improve health [32]. Previous research has shown that including exercise in interventions, enhances numerous health-related parameters independent of the effects on fatness. This includes lipid profile, blood-pressure and insulin sensitivity [33]. Interval exercise has been shown to be an effective and feasible type of exercise for overweight and obese individuals, due to large energy expenditure and relative short duration [34,35].

There is no doubt that overweight and obesity constitute a global health hazard, but there are widely differing opinions on how to reduce fat mass and improve cardiometabolic health. Primary prevention strategies include an adoption of healthy lifestyle behaviors [8], and studies have shown that treatments for CVD is more frequent in women than men in the primary prevention. Therefore, identifying an effective strategy for a healthy lifestyle that includes diet and exercise that promote weight-loss and improvement of markers of health, is a cornerstone for better health in females to prevent CVD. The need for weight reduction is of importance; however, enhancing body composition and improving CVD risk factors is also a priority to regain and maintain good health, as the cardioprotective effects of estrogen disappear after menopause, and the lipid profile aggravates [7]. In light of the positive effects of physical exercise and hypocaloric diets on health-related parameters in overweight and obese individuals, the aim of the study was to explore the effect of weight-loss achieved with calorie-restricted LCHF vs. normal diet only, or in combination with endurance exercise performed as interval exercise, on changes in health-related parameters. The secondary endpoints from the main study are presented in this paper, where we explore the effect on the LCHF diet and exercise on body composition, cardiorespiratory fitness and CVD risk factors.

## 2. Methods, Design and Setting

The study was a ten-week intervention where a 2 × 2 factorial design was used for a combination of diet and exercise. Participants were allocated randomly to one of four groups with a combination of normal or LCHF diet, with or without interval endurance exercise: normal-diet-only intervention group (NORM); LCHF-diet-only intervention group (LCHF); normal diet with exercise intervention group (NORM-EX); and LCHF with exercise intervention group (LCHF-EX) (Figure 1). The study was approved by the Regional Committee for Medical Research Ethics in Norway (2013/1529) and registered at www.clinicaltrials.gov as NCT04100356. The study was conducted in accordance with the Declaration of Helsinki and all participants provided written informed consent before entering the study. The study was conducted from October 2013 to April 2014 at The Norwegian School of Sport Sciences, Oslo, Norway and Atlantis Medical University College.

### 2.1. Participants

Participants were recruited through newspaper advertisements and social media. In total, 199 women showed interest in participating in the study and were screened for eligibility by e-mail and telephone interview. Eligibility criteria for the study were as follows: sedentary premenopausal Caucasian women, BMI 26.9–36.1, age 33–47 and living close to, or in Oslo. Exclusion criteria were pregnancy or breast-feeding, smoking or tobacco use, previous medical history of CVD, diabetes, endocrine disorder or kidney disease, and use of lipid-lowering or diabetes medication. After initial screening by email and telephone, 60 eligible participants were included. A study flow chart of enrollment and participant flow, as recommended by the Consolidated Standards of Reporting Trials (CONSORT) is presented in Figure 2.

### 2.2. Randomization

The research leader and assistant performed a simple online computer-generated randomization after baseline measurements (www.randomizer.org). Group allocation was sent to participants by email immediately after the randomization. Neither researchers nor participants were blinded to the intervention groups.

### 2.3. Intervention

Eligible subjects were allocated to four different interventions. The calorie deficit prescribed for all groups was −4900 kcal∙week^−1^ (−700 kcal∙day^−1^). This was achieved in the “diet-only” groups by reducing intake, and by a combination of exercise and reduction in intake in the “diet-exercise” groups. During exercise sessions, participants joined intervals until they reached an energy expenditure of 500 kcal∙session^−1^. The combination of treatments gave the following calculated deficit in the diet-only groups: NORM and LCHF (−4900 kcal∙week^−1^), and diet-exercise groups: NORM-EX and LCHF-EX (−3400 kcal∙week^−1^ deficit in intake, and −1500 kcal∙week^−1^ expenditure during exercise).

An introductory information meeting for the respective groups was held prior to beginning the study. A motivational speech was given to prepare the participants for the intervention and general questions were answered. Participants had coaching and follow-up from trained nutritionists (BA nutrition) twice a week, either by video meeting or phone call. Total energy expenditure (TEE) was estimated for each participant, using the Harris–Benedict equation [36] multiplied by the coefficient of Physical Activity Level (PAL) of each participant based on a 24-h activity log [37]. The log included activities such as sleeping, personal care, eating, cooking, sitting at work, housework, driving or public transportation, walking, and leisure activities.

### 2.4. Exercise Sessions

The exercise form was interval exercise for improving cardiovascular fitness, and to result in increased energy expenditure in the exercise groups. Participants in the two exercise groups attended an indoor bicycle exercise session three times a week (Mondays, Wednesdays and Thursdays), supervised by a qualified instructor. The exercise consisted of a 10 min warm-up, followed by 7 × 4-min intervals at a target intensity of 82–90% of HR_peak_ with a 3-min active rest at a comfortable recovering speed (~60% of HR_peak_). Approximately halfway through the session, participants were asked to rate their perceived exertion during exercise on the Borg scale [38]. Heart rate and energy expenditure during the session were recorded using Polar heart rate monitors (RS800CX, Polar Electro Oy, Kempele, Finland) [39]. Everybody participated in the interval session until they reached the goal of 500 kcal before cooling down for 5–10 min. Participants could not continue the interval session after reaching 500 kcal, and most participants performed seven intervals during each session. Participants in the exercise groups were asked not to participate in any other physical activities outside the study, whereas participants in the non-exercising groups were instructed to continue with their sedentary lifestyle.

### 2.5. Normal Diet

The diet recommended by the Norwegian Health Authorities [40], was used to determine and define the NORM diet in this study. The recommendations emphasize a diet composition of E% 10–20 protein, E% 25–40 fat and 45–60 E% carbohydrates (Figure 3). Intake of whole grains, legumes, fruits, berries, vegetables, vegetable oils, fish, shellfish, poultry, lean pork and beef, nuts, seeds, and restricted amounts of red meat was encouraged. Participants were advised to use vegetable oils or margarines for cooking, substitute whole-fat dairy with low-fat dairy products, avoid foods high in added sugar, tropical oils rich in saturated fatty acids, foods potentially rich in trans fatty acids and to avoid intake of alcohol.

### 2.6. LCHF Diet

The LCHF diet was a modified ketogenic Atkins diet [16,41], with an initial ketogenic phase. The increase in carbohydrates was done to increase the health-related quality of life and compliance [42] and in accordance with previous studies [16,43,44]. The combination of macronutrients in the LCHF groups restricted carbohydrate consumption to 20 g∙day^−1^ in the first week [16]. This equals approximately 5 E% carbohydrates. For the following weeks, the carbohydrate intake was increased by 10 g∙week^−1^ until participants reached a maximum 100 g∙day^−1^ of carbohydrates (Figure 3), similar to phase 4 in the Atkins diet [45]. The increase in carbohydrates throughout the intervention brought the participants gradually out of ketosis. Never the less, the carbohydrate content of the diet was still below the limits for the definition of the LCHF diet [46]. Fat intake was targeted at approximately 70 E% at the beginning of the trial. A proportional decrease in fat, alongside the increased carbohydrate intake, was planned throughout the first nine weeks of the trial. The protein intake was targeted at 25 E% throughout the trial [32]. Consumption of meat, fish, shellfish, poultry, nuts, eggs, seeds, whole-fat dairy products, berries, vegetable oils (including fruit-, nut-, seed- and tropical oils) and vegetables and legumes, low in carbohydrates, was emphasized in the first weeks. Food high in unsaturated fatty acids was encouraged. Participants were advised to avoid foods with added sugar and potentially rich in trans fatty acids, and to avoid intake of alcohol. Vegetables, fruits and whole grains were suggested as good sources of carbohydrates when the targeted amount increased.

### 2.7. Study Procedures

Baseline data were collected during an approximately three-week run-in. All tests were repeated post-intervention after the ten-week intervention, in primo April (Figure 4). Weight was measured every second week throughout the intervention to assess weight loss. Participants attended the laboratory in the afternoon, weight was measured on a Bioelectrical Impedance Analysis (BIA) device in a non-fasting state, wearing light clothing. The weight of light clothing was estimated to 0.5 kg and preregistered in the BIA device, which subtracts the clothing from the participants’ weight. Habitual dietary intake served as the baseline and was assessed the last week prior to the intervention; participants were instructed to weigh their food and beverages and register three weekdays and one weekend-day. During the 10-week intervention, dietary records were kept every day throughout the study and controlled by nutritionists.

## 3. Outcome Measurements—Clinical and Laboratory Procedures

### 3.1. Anthropometric Measurements

Height was measured to the nearest 0.5 cm, on a standard wall-mounted stadiometer (Seca 206 Stadiometer Wall Mounted, Seca, Deutschland, Hamburg, Germany). Waist and hip measurements were performed with a medical non-stretchable measurement tape and measured to the nearest 0.1 cm. Waist circumference was measured halfway between the lower point of the rib arch and the top of the iliac crest, parallel to the floor at the end of a normal expiration. Hip measurement was done at the fullest part of the hip. Measurements were repeated three times and the mean was calculated. Weight was measured on a calibrated bioelectric impedance device (MC 180 MA Multi Frequency, Tanita, Tokyo, Japan) (BIA).

### 3.2. Body Composition

A whole-body scan was performed, using Dual-energy X-ray absorptiometry (DXA) (Lunar iDXA, GE Healthcare, Madison, WI, USA) and analyzed by enCORE (software version 14.10.022). The body scan was done in the fasted state in the morning, wearing underwear, without any metal objects or shoes.

### 3.3. $\dot{V}$O_2peak_ and HR_peak_

Testing of peak oxygen uptake ($\dot{V}$O_2peak_) was performed in the afternoon, in a fed state. The testing was completed using an incremental test on an ergometer bicycle (Excalibur Sport Cycle Ergometer, Lode, The Netherlands). Oxygen consumption and carbon dioxide production were measured using an automatic O_2_/CO_2_ analyzer (Moxus Modular Metabolic System, AEI Technologies, Inc.) with a breath-by-breath average calculated in 30-s intervals throughout the protocol. A two-point gas calibration, in addition to volume calibration, was done daily prior to testing. A 5-min warm-up was performed prior to the $\dot{V}$O_2peak_ test, which started at 50 W with an increase of 15 W every 30 s until exhaustion. Oxygen uptake was measured throughout the protocol. Capillary blood was sampled one minute after termination of the $\dot{V}$O_2peak_ test and analyzed for lactate (Akray, KDK Corporation, Kyoto, Japan). An increase of less than 1 mL∙kg∙min^−1^ after two increments in workload, combined with a respiratory exchange ratio (RER) > 1.10 and lactate above 7 mmol∙L^−1^, was set as the criterion for $\dot{V}$O_2peak_. Peak heart rate (HR_peak_) was recorded (RS800CX, Polar Electro Oy, Finland) during the $\dot{V}$O_2peak_ test and noted as the highest heart rate.

### 3.4. Lipids

Blood samples for total cholesterol, LDL, HDL and TG were collected after a 12-h fast, 36 h after last the exercise session. Samples were collected in serum separator tubes (Vacutainer SST 8.5 mL, BD, Franklin Lakes, NJ, USA) and coagulated for 30 min at room temperature before centrifugation (Eppendorf 5072R, Hamburg, Germany). Samples were stored at 4 °C for three hours before analysis at Fürst Laboratory, Oslo, Norway (Advia Centaur XPT, Siemens Medical Solutions Diagnostics, Tokyo, Japan).

### 3.5. Dietary Assessment

Food was self-prepared and weighed on an electronic scale (1 g precision) during the intervention. Dietary intake was registered in an online dietary registration program (www.somebody.no, Somebody AS © 2008–2016), and total energy intake, carbohydrate, fat and protein intake were calculated. This specific tool was chosen as it is user-friendly. This was important, as participants were to use it daily. Participants were provided with an online account for the entire intervention period. A total of six nutritionists supervised the participants. The nutritionists supervised participants from different groups in order to reduce bias due to subjective diet guidance. Nutritionists had access to their clients’ accounts and registered the individual energy targets. Nutritionists assessed food registration and gave advice and suggestions for food and drink according to the respective group during the intervention. Clear dietary targets were provided with individual macronutrient targets for each participant. High levels of encouragement were given by using motivational techniques including goal setting, feedback on weight loss and dietary compliance achievements [47]. Each participant had two motivational sessions per week with a nutritionist. The motivational sessions were held on Skype or by telephone once a week, and at the preference of the participant (Skype, telephone or mail) at a second time during the week. Closed groups on social media were created for each group and administered by the project leader and nutritionists. The groups were created to let participants communicate and share information and recipes, troubleshoot common nutritional issues and generally increase motivation and compliance. A standard operating procedure (SOP) was followed in order to carry out the guidance in a similar way and achieve weight loss.

To monitor ketosis and compliance in the LCHF groups, acetoacetate was estimated in morning urine with urine sticks (Ketostix 2880, Bayer, Berlin, Germany) from day one of the LCHF diet. The sticks change color depending on the level of ketones found in the urine. Participants compared the strip to a color chart on back of the box, with a scale reading trace (0.5 mmol∙L^−1^), small (1.5 mmol∙L^−1^), moderate (4 mmol∙L^−1^) and large (8–16 mmol∙L^−1^) defined by the manufacturer. Participants registered the results in the dietary registry program, where nutritionists could control diet compliance.

## 4. Sample Size Calculation and Statistical Analysis

The results in this paper are from a larger study where the primary endpoint was glucose tolerance (manuscript in draft). Sample size was calculated using an online calculator (http://www.math.yorku.ca/SCS/Online/power/). Sample size was based on the effect of exercise and weight loss on glucose tolerance [25,48,49]. Impaired glucose tolerance is correlated to disturbance in lipid metabolism, which increases the risk for CVD [50]. As both weight reduction and exercise can improve glucose tolerance, we anticipated a 19% drop in AUC glucose during 2 h OGTT after weight loss with, or without exercise. With an estimated difference in AUC glucose between groups of 150 U and an SD of 130 U, a sample size of 12 participants was needed for each group, with a power of 80% at a two-tailed significance level of 0.05. With an assumption of 15% dropout, we aimed to recruit 15 participants in each group.

Main analyses of outcome variables (body composition, cardiorespiratory fitness and CVD risk factors) were performed with linear mixed models to assess the differences between groups after the intervention. The models included group, time, and group*time interaction set as fixed variables. Measurements were set nested within subject and time was included as a random slope if it improved the model, which was evaluated with a likelihood ratio test. Analyses followed the intent to treat principle, with last value measured included for dropouts. We completed pairwise comparisons within (pre-post) and between comparable groups. Results for both within and between groups are presented in the result section. Differences within groups at post-measurement were adjusted for baseline measurements. Due to multiple comparisons, all pairwise comparisons were also assessed with Bonferroni adjustments. Assumptions were examined with visual inspections of residuals and model assumptions were considered met. Descriptive analysis and differences between groups were assessed using a *t*-test with unequal variances for continuous variables. The variables included age, energy intake, energy deficit, dietary nutrient content, baseline nutrient intake, energy expenditure, exercise intensity, kcal expenditure and RPM during exercise sessions, and attendance to exercise sessions. Missing values for baseline measurements of saturated fats were imputed by mean imputation. Analyses were completed in Stata version 15.1 software (StataCorp. 2017. Stata Statistical Software: Release 15. College Station, TX, USA: StataCorp LCC) with two-sided *p*-values and significance level set to 5%.

## 5. Results

### 5.1. Study Participants

A total of 60 women were eligible for participation, but three women withdrew during the baseline measurements. The study included 57 Caucasian premenopausal, overweight women, aged 33–47 years, who were randomized into four different intervention groups. The dropout rate was 7% (*n* = 4) during the intervention. One participant withdrew due to her work situation, two gave no reason for withdrawal and one participant did not adhere to the diet protocol and dropped out (Figure 1). Recruitment was between October 2013 and January 2014 with first participant enrolled primo October 2013. Baseline measurements were performed in January 2014, and the intervention was conducted from 27 January to 7 April 2014 (Figure 4). Baseline characteristics for each group are shown in Table 1.

### 5.2. Results/Study Outcomes

#### 5.2.1. Weight

The energy deficit resulted in weight loss in all four groups (within-group), with no differences between groups after the intervention. All four groups achieved a weight loss > 5% of the initial weight. The NORM group lost 5.2 ± 2.3 kg, whereas the other diet-only group LCHF lost 6.2 ± 2.1. The exercise groups, NORM-EX and LCHF-EX lost 5.5 ± 2.2 kg and 6.7 ± 2.3 kg, respectively. Weight loss at six weeks was similar in all groups with no differences between the groups; NORM 4.1± 1.7 kg, LCHF 5.0 ± 1.9 kg, NORM-EX 3.7 ± 1.7 kg and LCHF-EX 4.6 ± 1.6 kg.

#### 5.2.2. Body Composition

The intervention resulted in a loss of fat mass (FM) in all groups, but no between-group differences were seen (Table 2). The mean reduction for pooled groups was 13%, with the NORM group showing the least loss of FM (8%, *p* < 0.001) and the LCHF group showing the largest loss (17%, *p* < 0.001).

In regard to visceral fat, there was a reduction in all groups in response to the intervention, where the mean reduction was 24%, but no between-group differences were observed. The NORM group had the least reduction in visceral fat, of 9.1% (*p* < 0.001), where the other diet-only group, LCHF, had a large reduction of 33% (*p* < 0.001).

All groups except the NORM group achieved a reduction in % fat. The LCHF group had the greatest reduction of 5.9% (*p* < 0.001). Between-group comparison showed a difference in % fat comparing NORM with NORM-EX (*p* = 0.024), where NORM-EX had a large reduction, and the NORM group had an increase after the intervention.

All groups showed a within-group reduction in LBM, with a mean reduction of 2.7% for all groups pooled, but no between-group differences were seen.

#### 5.2.3. Waist-to-Hip Ratio

No difference in waist-to-hip ratio was observed when comparing groups after the intervention (Table 2). The NORM-EX and LCHF groups showed a 3.5% decrease (*p* = 0.002) and the NORM group showed a 2.3% decrease (*p* = 0.044). The decrease in the LCHF-EX group was 2.3% but did not reach statistical significance (*p* = 0.067).

#### 5.2.4. Cardiorespiratory Fitness

Both exercise groups showed a within-group improvement in cardiorespiratory fitness in response to the intervention (Table 3). The LCHF-Ex group had a 6.9% increase (*p* < 0.001) and the NORM-EX group had a 9.6% increase (*p* < 0.001) in $\dot{V}$O_2peak_. In contrast to the exercise groups, the diet-only groups showed a 5.3% (NORM *p* < 0.01) and 5.4% (LCHF *p* < 0.001) reduction in $\dot{V}$O_2peak_ in L∙min^−1^ (*within-group*). Between-group comparison showed a difference comparing the NORM-EX group with the NORM group (*p* = 0.001) after the intervention, due to an increase in the NORM-EX whilst the NORM group had a reduction in response to the intervention.

#### 5.2.5. Lipid Profile

The intervention resulted in a reduction in total cholesterol in all groups (Table 3). Between-group comparisons showed that the NORM-Ex group reached lower levels in total cholesterol, compared to the NORM group (*p* = 0.003) and the LCHF-EX group (*p* = 0.004) after the intervention.

Within-group comparison showed that all groups except the LCHF group had a decrease in LDL after the 10-week intervention (NORM *p* = 0.005, NORM-EX *p* = <0.001, LCHF-EX *p* = 0.001, Table 3). The intervention also resulted in between-group differences in LDL cholesterol, as the NORM-EX group reached a lower LDL cholesterol level than the LCHF-EX group (*p* = 0.018).

Within-group comparison showed that all groups had a reduction in HDL cholesterol in response to the intervention: NORM 19%, LCHF 13%, NORM-EX 21%, LCHF-EX 13%, (*p* < 0.001 all groups, Table 3). In addition, HDL levels in the NORM-EX group were reduced to 1.1 mmol∙L^−1^, which is below the target for primary prevention. The intervention resulted in an HDL difference between the groups after the intervention, where the NORM-EX group decreased to lower levels than the NORM group (*p* = 0.029).

Within-group comparison showed that the two exercise groups achieved a reduction in TG (Table 3). The LCHF-EX group had a 29% decrease (*p* = <0.001) and the NORM-EX had a 37% decrease (*p* = <0.001). No changes were seen in the NORM and LCHF groups. Between-group differences were seen between the NORM-EX and NORM groups after the intervention (*p* = 0.030).

The diet-only groups both showed an increase in the total cholesterol/HDL ratio, where the NORM group had a 9.0% (*p* = 0.007) increase and the LCHF group had an 8.8% increase (*p* = 0.004). No difference was seen between-groups (Table 3).

## 6. Ancillary Analyses

### 6.1. Energy Deficit—Diet Only or Diet and Exercise

Within-group change showed that all groups reduced their energy intake during the intervention (Table 4). The energy deficit from estimated requirement was −893 kcal∙day^−1^ in the NORM group (37%), −901 kcal∙day^−1^ in the LCHF group (37%), −829 kcal∙day^−1^ in the NORM-EX group (42%) and −796 kcal∙day^−1^ in the LCHF-EX group (43%). No difference was seen between-groups in energy intake during the intervention. When combining restriction in diet and adding exercise expenditure, the exercise groups had a larger calculated deficit per day than comparable diet-only groups. The LCHF-EX had a total deficit of −1003 kcal∙day^−1^ and LCHF had −901 kcal∙day^−1^ (*p* = 0.006), whereas the NORM-EX group had a total deficit of -1029 kcal∙day^−1^, and the NORM group had a deficit of −893 kcal∙day^−1^ (*p* = 0.041). No differences were seen between the diet-only groups, or when diet + exercise groups were compared.

**Table 4 nutrients-13-00110-t004:** Energy requirements, expenditure, intake and macronutrients at baseline and during the intervention.

	NORM (*n* = 15)	LCHF (*n* = 14)	NORM-EX (*n* = 14)	LCHF-EX (*n* = 14)	NORM vs. LCHF	NORM vs. NORM-EX	LCHF vs. LCHF-EX	NROM-EX vs. LCHF-EX
Nutrition					Between Group Differences at Baseline vs. Post Intervention (*p*-Values) ^			
Energy								
Requirement (BMR*PAL)	2437 ± 171	2440 ± 192	2430 ± 188	2337 ± 153	0.963	0.926	0.152	0.199
Intake (kcal) pre	2486 ± 173	2489 ± 195	2488 ± 183	2400 ± 150	0.969	0.974	0.197	0.177
Intake (kcal) post	1544 ± 124	1539 ± 212	1601 ± 244	1541 ± 126	0.927	0.405	0.892	0.470
Within-group change (*p*-value)	**<0.001**	**<0.001**	**<0.001**	**<0.001**				
Energy availability *	1544 ± 124	1539 ± 212	1401 ± 237	1334 ± 89	0.942	0.041	0.006	0.413
Macronutrients								
Carbohydrate (E%) pre	48.7 ± 3.1	47.7 ± 3.3	50.2 ± 3.3	48.1 ± 2.3	0.425	0.205	0.695	0.067
Carbohydrate (E%) week 1-10	48.4 ± 2.8	14.0 ± 2.9	50.0 ± 3.0	15.0 ± 1.2	**<0.0001**	0.929	0.341	**<0.0001**
Within-group change (*p*-value)	0.809	**<0.001**	0.978	**<0.001**				
Protein (E%) pre	14.9 ± 1.5	14.8 ± 1.1	15.0 ± 1.0	15.2 ± 2.2	0.845	0.929	0.547	0.709
Protein (E%) week 1-10	20.2 ± 1.9	24.2 ± 1.4	19.2 ± 1.8	24.0 ± 0.9	**<0.0001**	0.174	0.656	**<0.0001**
Within-group change (*p*-value)	**<0.001**	**<0.001**	**<0.001**	**<0.001**				
Fat (E%) pre	36.5 ± 3.2	37.5 ± 3.2	34.8 ± 3.1	36.6 ± 3.2	0.385	0.164	0.476	0.133
Fat (E%) week 1–10	31.4 ± 2.7	61.9 ± 3.9	30.8 ± 3.4	61.0 ± 1.2	**<0.0001**	0.641	0.497	**<0.0001**
Within-group change (*p*-value)	**<0.001**	**<0.001**	**0.020**	**<0.001**				
Saturated fat (E%) pre	9.9 ± 2.6	11.0 ± 2.2	10.1 ± 3.1	11.9 ± 3.5	0.249	0.809	0.412	0.168
Saturated fat (E%) week 1–10	7.5 ± 1.7	13.8 ± 3.5	7.6 ± 1.7	12.8 ± 2.8	**<0.0001**	0.826	0.436	**<0.0001**
Within-group change (*p*-value)	**0.005**	**0.010**	**0.014**	0.562				

Data are presented as mean ± standard deviation (SD). Linear mixed models, with pairwise comparisons were used to evaluate within- and between-group differences. Significant values are presented with bold numbers. ^ Differences between groups at post-measurements are adjusted for baseline measurements. NORM: normal diet, LCHF: low-carbohydrate high-fat diet, NORM-EX: normal diet combined with exercise, LCHF-EX: low-carbohydrate high-fat diet combined with exercise. * Energy expenditure during bicycle sessions has been subtracted and is presented in Table 5.

### 6.2. Macronutrient Intake

#### 6.2.1. Carbohydrate

As expected, a reduction in E% carbohydrate was observed in the LCHF groups during the intervention (Table 4). The average carbohydrate reduction for both LCHF groups was 70% (*p* <0.001). No significant change in E% carbohydrate was observed in the NORM groups. Between-group differences were seen when comparing NORM vs. LCHF (*p* < 0.0001) and NORM-EX vs. LCHF-EX (*p* < 0.0001) after the intervention.

#### 6.2.2. Protein

Within-group changes were seen in all groups, where all groups increased their E% intake of proteins (Table 4). The E% for protein was increased by 32% in NORM groups (*p* < 0.001), and 61% in the LCHF groups (*p* < 0.001).

Between-group differences were seen after the intervention when comparing NORM vs. LCHF (*p* < 0.001) and NORM-EX vs. LCHF-EX (*p* < 0.001).

When the protein intake in grams was divided by kilograms body weight and kilograms LBM there was a significant higher intake in the LCHF groups compared with the NORM groups; LCHF vs. NORM (*p* < 0.001) and LCHF-EX vs. NORM-EX (*p* < 0.001) (Table 6).

#### 6.2.3. Fat and Saturated Fat

During the intervention, a within-group increase was seen in E% fat in the LCHF groups pooled as they increased by 66% (*p* < 0.001, Table 4). A within-group decrease was seen in the NORM groups pooled by 13% (*p* < 0.001). Between-group differences were seen when comparing NORM vs. LCHF diet-only groups, with a higher E% fat in the LCHF group than in the NORM group (*p* < 0.0001, Table 4). A similar difference was seen between the LCHF-EX group and the NORM-EX group (*p* < 0.0001). Changes in E% saturated fat were observed in three of the groups after the intervention (within-group); the NORM group had a 25% reduction, and the LCHF group had a 25.5% increase. No change was seen in the LCHF-EX group. Between-group differences were seen when comparing LCHF groups with their respective NORM groups. During the intervention, the LCHF group had a higher E% intake of saturated fat compared with the NORM group (*p* < 0.0001), and the LCHF-EX group had a higher intake than the NORM-EX group (*p* < 0.0001, Table 4).

### 6.3. Ketosis

The carbohydrate restriction resulted in an increase in acetoacetate (trace 0.5 mmol∙L^−1^) from Day 2 in participants in the LCHF groups. Ketosis gradually declined when the amount of carbohydrate was increased each week throughout the study. Ten participants from each LCHF group reported that ketosis results in the dietary register program. The participants in the LCHF-EX group had on average 49 days out of 70 in ketosis and the LCHF group had 37 days out of 70. The number of participants in ketosis declined gradually throughout the intervention. The estimated urine ketone levels during the period spent in ketosis for the LCHF-EX and LCHF groups were 3.4 ± 2.8 mmol∙L^−1^ (*n* = 10) and 2.1 ± 1.3 mmol∙L^−1^ (*n* = 10) respectively.

### 6.4. Exercise Compliance, Energy Expenditure and Exercise Intensity

Training attendance was high. The NORM-EX group had a mean attendance of 88 ± 7% and the LCHF-EX group had a mean attendance of 93 ± 7% (Table 5). In addition, all participants reached a goal of 500 kcal expenditure during each exercise sessions (Table 5). Participants that missed out on an exercise session got their energy intake adjusted by nutritionists to balance for the inadequacy of expenditure.

Average energy expenditure during each indoor bicycle session was similar in the two exercise groups (*p* = 0.373, Table 5). Exercise intensity measured as % HR_peak_ did not differ between the exercise groups during the intervention (*p* = 0.630, Table 5). The rate of perceived exertion (RPE), estimated using the Borg scale, was similar throughout the intervention and did not differ between the groups (*p* = 0.94, Table 5).

## 7. Unintended Effects

No serious, harmful or unintended effects were reported. Minor non-serious and well-known side effects of the LCHF diet were reported, such as dizziness (*n* = 19), mild headache (*n* = 12) and lack of power during bicycle sessions (*n* = 8) during the first two weeks.

## 8. Discussion

The present study was designed to compare the effect of weight loss, achieved with two different diets with and without exercise, on glucose tolerance (manuscript in draft), body composition and CVD risk factors in overweight and obese females. A desirable weight loss of >5% was achieved in all groups with the prescribed calorie deficit. Previous results have shown that a modest weight-loss as low as 5% can reduce and eliminate metabolic disorders associated with overweight and obesity [13]. The weight loss the participants achieved was in accordance with the study design, which involved a prescribed kcal reduction of −4900 kcal∙week^−1^ and use of SOP to increase compliance to weight loss protocol. Similar weight loss as a result of kcal deficit has previously been reported in both diet-only groups [15], and in diet combined with exercise [27]. The effect of weight-loss on cardiometabolic health seen in this study is reflected in the improvements in most of the CVD risk factors studied. These improvements were achieved even with the lack of change in %fat seen in the NORM group.

### 8.1. Body Composition

The current study confirms the effect of calorie deficit on reduction in FM (kg), as all our groups achieved a reduction in FM in response to the 10-week intervention. Although there is a functional diversity of adipose tissue, there is no doubt that total body fat is a major contributor to the metabolic challenges often seen in overweight and obesity and increases the risk of CVD. Weight loss improves metabolic health, but only when fat mass is reduced, and reduction in FM has been correlated with improved health and reduction in metabolic disorders [5]. Body fat distribution is associated with CVD risk, and visceral fat is associated with more adverse CVD risk profiles [2]. We observed a large reduction in visceral fat in all our participants, and these results are in line with other weight-loss studies [5,25,51,52]. It has been shown that, during weight loss, visceral fat is more responsive to lipoprotein lipase than subcutaneous adipose tissue, and that visceral fat is more affected percentage-wise than subcutaneous fat [2]. Even though the beneficial effect of exercise in reducing visceral fat is well documented [25,53], we did not observe any differences between the groups, neither in FM nor visceral fat as previously reported by others [25,54].

We observed a loss in LBM (kg) in all groups, which is identified as an adverse effect of weight loss [32]. The role of LBM as an energy-demanding tissue is important [30] and conservation of LBM during weight loss can be essential in maintaining the resting metabolic rate [31,55]. Exercise can diminish the reduction of muscle mass during weight loss, independent of diet [27,56,57]. In contrast to previous studies, we observed neither a muscle-sparing effect, nor an increase in LBM in our exercise groups, although a relatively large mass of muscle was used during exercise sessions. The muscle sparing effect is more pronounced in studies with resistance training [57]; however, it is also seen in studies with interval endurance exercise using the leg muscles [29], which account for a large percentage of total muscle mass, albeit dependent on body shape and fitness. All groups had protein content >0.8 g∙kg^−1^ (Table 6) which is considered the minimum amount to avoid reduction of LBM [58]. Previous research has reported the need for a protein content of 1.8 g∙kg^−1^ to help offset loss in LBM [27], whereas Hansen et al. found no relationship between the amount of protein ingested and loss of LBM, as long as a minimum of 1 g∙LBM^−1^ was provided [59]. All our groups had a higher intake than 1 g∙LBM^−1^ during the intervention (Table 6). A possible explanation for the loss of LBM in our study may be that the energy deficit in the intervention was high, and such a large deficit decreases the postprandial rate of muscle protein synthesis, increases the catabolic state and enhances an unfavorable reduction in LBM [58]. There was no difference between groups when comparing loss of LBM.

Both exercise groups, as well as the LCHF group, achieved favorable changes in % fat. The % fat did not change in the NORM group (without exercise), despite a significant weight reduction. This finding is unexpected, as previous weight-loss research has shown a reduction in primarily FM, although combined with some reduction in LBM [27,58,59]. The lack of change in % fat in the NORM groups is explained by the parallel large loss of LBM (kg) together with FM (kg) loss, which affects the body composition and % fat. The large loss of LBM is considered quite negative, although inevitable during large calorie deficit and has negative effect on the % fat.

The lack of improvement in % fat in the NORM group in response to the intervention, resulted in a between-group difference for the NORM and NORM-EX groups after the intervention. However, the reason behind this absence of reduction in % fat in the NORM group can only be speculated upon at this time, as the combination of macronutrients, calorie deficit and the amount of protein g∙kg^−1^ was similar in the two groups. However, this supports the recommendation that participants in weight-loss programs should be encouraged to participate in regular exercise.

### 8.2. Cardiorespriatory Fitness

Both exercise groups improved their oxygen uptake markedly in response to the intervention and it is noteworthy that all participants in the exercise groups achieved an increase in $\dot{V}$O_2peak_. The increase seen in the LCHF-EX group is similar to results from a study done on overweight men [60], whereas the increase in the NORM-EX group was slightly higher than that seen in this comparable study. It is noteworthy that the increase we observed in the NORM-EX group is the same as previously seen in young, normal weight, healthy subjects after an 8-week intervention with similar exercise frequency and intensity [61]. The observed increase in $\dot{V}$O_2peak_ confirms the positive effect of exercise on cardiovascular fitness and the results are in line with improvements reported in other studies [62]. The goal for each session was to achieve an energy expenditure of 500 kcal, where the participants followed a controlled interval exercise program performed as 4-min intervals, which resulted in an average intensity of ~80% of HR_peak_ (vigorous activity) during exercise sessions, with 53% of the exercise time at 80% HR_peak_ or above. It has been shown that interval exercise at higher intensities is superior to continuous exercise to gain improvements in $\dot{V}$O_2peak_ [62]. The intensity during exercise sessions in the intervention was well above recommendations given by the Health Authorities in Norway [40]. Our results support previous research that shows that increased physical activity should be one of the factors implemented in the quest for a healthier lifestyle. As expected, no improvements were seen in cardiorespiratory fitness in the non-exercising groups.

### 8.3. LDL

Three of the groups achieved improvements in their LDL levels, whereas the LCHF group did not. Although weight loss is a recommended method to reduce LDL levels in overweight and obese people [4], the inadequacy of LDL reduction in the LCHF group despite the weight loss can be explained by the higher intake of saturated fatty acids during the LCHF diet as predetermined by protocol (Table 3). Previous LCHF studies with weight loss have resulted in increased LDL at six weeks [63] and three months [16]. Our recent study on normal-weight young women has shown that an LCHF diet high in saturated fat increased LDL levels in this group to levels above the target for primary prevention in only three weeks [17]. These results are supported by meta-analyses that concluded that LCHF diets, even with a pronounced weight-loss, resulted in an increase in LDL [15].

In contrast, studies with overweight participants have shown improvement in the LDL profile after an LCHF diet, where the participants already had high LDL levels and a distorted lipid profile prior to the study [64]. Both our LCHF groups had LDL levels above the target for primary prevention prior to the intervention, but only the LCHF-EX group achieved improvement. Our results are not in line with previous results, where improvements could have been expected due to higher LDL levels in our participants at baseline. Studies have shown that exercise training and volume are an important factor for CVD risk improvement, where a large volume of exercise is related to a decrease in the dense subfraction of LDL [65]. Although exercise has been shown to improve lipid profiles even in the absence of weight loss, a number of studies have shown that LDL levels are not improved with aerobic exercise unless weight loss occurs [24]. As a result of the intervention, the effects of exercise in combination with weight loss seems to have counteracted the negative effect of the diet in the LCHF-EX group (within-group change), although a between group comparison showed no difference between the LCHF and LCHF-EX groups. The between-group comparison showed a significant improvement in LDL in the NORM-EX group compared to the LCHF-EX group after the intervention. This result was not expected as the combination of weight-loss and exercise was expected to counteract the effect of higher intake of fats, including saturated fats, in the LCHF groups (Table 4) [17,63]. It has to be taken into consideration that there is an ongoing debate as to whether the size of the LDL particles is of importance in regard to CVD risk and it has been proposed that risk prediction may be improved by using information on LDL particle number and size [66]. Studies on the effect of exercise without weight loss have previously demonstrated that the amount of larger LDL particles is increased relative to smaller dense particles [65], but the effect of the combination of an LCHF diet and exercise on LDL particle size needs further exploration. However, the improvements in LDL seen in the NORM-EX groups speaks for this combination during weight-loss programs that include diet and exercise.

### 8.4. HDL

The reduction in HDL seen in this study has been demonstrated in previous highly controlled weight-loss studies, where both non-HDL cholesterol and HDL were reduced [67]. Our results are in line with results from a highly controlled weight-loss study, although that study was of much shorter duration [67]. The reduced HDL during weight-loss is probably a response to a low turnover of endogenous and exogenous VLDL. This triglyceride-rich lipoprotein will be reduced as the TG synthesis is turned down as a result of a decrease in lipoprotein lipase during the hypocaloric state, and the need for mobilization of energy stored as fat [68]. Therefore, the reduction in HDL seen in our study is a natural response due to changes in metabolism. In our study, we did not observe an increase in HDL in either exercise groups, and we observed a greater reduction in HDL in the NORM-EX group compared to the NORM group. This is in contrast to other studies [24]. One possible explanation is the positive effect of exercise on adipose tissue mobilization, where hormone-sensitive lipase plays an important role, and a training-induced increase in fat oxidation and decreased demand for reversed cholesterol transport. The combination of these factors can augment the decrease in HDL in the NORM-EX group, which actually fell below the levels for primary prevention. The aforementioned reduction in lipoprotein lipase during energy restriction calls for the mobilization of lipids from adipose tissue to serve as fuel in both groups. This results in limited transfer of TG to HDL and causes lower levels of the protective HDL, as previous studies have shown that HDL seemed to increase after a period of weight loss, where an HDL increase of 0.009 mmol∙L^−1^ per kg lost after entering a weight-stable condition was observed [68]. As HDL does not only serve as a transporter for lipids, but is also a potent antioxidant, has anti-inflammatory and anti-thrombotic functions [69], the reduction seen in the NORM-EX group is of concern. On the other hand, as exercise has been shown to increase HDL [24], the inclusion of exercise in weight-loss programs and beyond might be of higher importance than we realize. However, the significant difference between the NORM and NORM-EX groups in disfavor of the NORM-EX group cannot be overlooked.

### 8.5. Total Cholesterol/HDL Ratio

Within-group changes showed a decrement in total cholesterol in both NORM and LCHF groups; however, the HDL levels decreased concurrently. This resulted in no significant changes in the total cholesterol/HDL ratio, and levels that were still above the optimal level (<3.0) after the intervention. An increase was seen in the total cholesterol/HDL ratio in the NORM and LCHF groups, which indicates an increment in the CVD risk [70]. The augmentation should be noted as an adverse metabolic change that can become more prominent with age, where the CVD risk increment will become more distinct. As there was no difference between groups, we cannot draw a conclusion about the optimal combination of diet and exercise.

### 8.6. Triglycerides

The exercise groups achieved a large reduction in TG in response to the intervention, while the diet-only groups did not show changes in TG levels. Weight loss has been associated with an improvement in lipid profile in overweight and obese individuals with a decrease in TG [11,12], and weight loss mediated by LCHF diets has been superior to normal diets in TG reduction [15,64]. This did not seem to be the case in our participants, as the LCHF group did not achieve a significant improvement. The reason may be that the LCHF group had TG levels within the normal range at baseline (<1.7 mmol∙L^−1^), and further improvement can be difficult to achieve during such a short intervention. For comparison, the LCHF-EX group showed a reduction in TG in response to the intervention. This difference in responses in the LCHF groups was not significant when comparing those groups. Although within-group comparison showed the positive effect of exercise, we cannot conclude with exercise as superior to LCHF diet only when it comes to improvements in TG. However, the effect of the intervention in the LCHF-EX group is in line with results from a meta-analysis of randomized controlled trials that reported an 11% reduction in TG after aerobic training in obese and overweight participants with a modest weight loss [71]. Further on, we observed larger improvement in TG levels in the NORM-EX group than in the NORM group, which again highlights previous exercise studies that have shown the positive effect of exercise on TG levels. Studies have also shown that high-intensity interval exercise is superior to moderate continuous exercise, measured as a better postprandial lipemia with a high TG clearance rate [72]. Although the best timing of TG measurements is debated in regard to the positive effect of exercise on TG metabolism [24], our results support previous findings and emphasize the importance of acute- and post-exercise effects on TG clearance, notably not in favor of either NORM-EX or LCHF-EX.

### 8.7. Energy Deficit and Diet Compliance

The total energy deficit recommended for all groups for the 10-week intervention was 49,000 kcal. This deficit equals weight loss of 5.5 kg FM, which is 6.2% of participants’ average weight at baseline, and within the desired weight loss of >5%. Comparing the total energy deficit reported with weight loss achieved, can be used to estimate participants’ compliance to the diet. This comparison shows that the groups in the study possibly underreported intake, ingested less than reported and/or increased expenditure. The NORM group had a total deficit of 62,510 kcal during the 10 weeks, the LCHF group 63,070 kcal, the NORM-EX 72,030 kcal and the LCHF-EX had a deficit of 70,210 kcal in total. The calculations estimate that the NORM group underreported 339 kcal∙day^−1^, the LCHF group underreported 93 kcal∙day^−1^, the NORM-EX group underreported 368 kcal∙day^−1^, whereas the LCHF-EX group correctly reported kcal∙day^−1^, as estimated weight loss in regard to kcal reported was the same as achieved weight loss. However, these calculations have to be interpreted with care and can only give an indication of compliance to the diet. First, the prescribed calorie intake was calculated from kcal requirements estimated from BMR (the Harris–Benedict equation produced by DXA), multiplied by the PAL coefficient for each participant, based on a 24 h individual log (Table 4). The fact that many individuals overestimate their physical activity [73] could have caused higher estimated requirements in the beginning of the trial, which in turn would result in false underreporting. Second, the participants were not equipped with activity trackers to measure daily activity. Third, we have calculated the weight loss as FM, where in reality weight loss is both LBM and FM, and the amounts of kcal stored in each tissue is unequal [74]. Nevertheless, even with underreporting intake, which is common in free-living participants [75], weight loss in all groups was above the goal of >5%.

## 9. Strength and Limitations

The study included only females, which is a strength of the study. Females and CVD risk factors are not studied to the same extent as CVD risk factors in males, albeit the physiology [76] and the effects of diet [77] and exercise [77,78] on weight-loss and CVD risk factors are different when comparing the sexes. Another strength of the current study is participants’ daily registration of food and beverage intake, and the close and frequent follow-up by nutritionists throughout the intervention. Lastly the tightly supervised and controlled exercise program with regard to both intensity and energy expenditure is also a strength of the study, in combination with a high compliance with the exercise program.

A limitation of the study is a small sample size. Another limitation is self-reported food consumption, where it is generally known that under-reporting is common. At the same time, participants’ over-estimation of physical activity (PAL calculation) at the beginning of the trial may have delayed the weight loss by a couple of weeks, and therefore a longer trial may be necessary to detect differences between groups. Registration or control for any additional activity was also a limitation in our study. Further on, it is also important to consider whether the intervention arms received similar intensity and attention, as the exercise groups met three times weekly during exercise, where diet-only groups did not have a similar program and communicated mainly through social media.

## 10. Conclusions

### 10.1. Body Composition

A 10-week weight loss program with exercise and/or LCHF diet showed improvements in different CVD risk factors in all groups, while weight loss achieved with exercise and diet produced better overall results than a normal diet only. Between-group comparison showed that the normal diet-only did not induce a loss in % fat tissue in response to the intervention. A normal diet in combination with exercise is superior to a normal diet-only when it comes to reduction in % fat tissue.

### 10.2. Cardiorespiratory Fitness and Cardiovascular Risk Factors

The intervention showed that exercise, independent of diet, has beneficial effects on cardiorespiratory fitness, showing between-group differences between NORM-EX and NORM. A normal diet in combination with exercise caused greater improvement in total cholesterol, LDL and TG compared to the other groups, but also the largest decrease in HDL. The concurrent decrease in HDL in all groups stalled improvements in the total cholesterol/HDL ratio. As the NORM-EX group had the highest reported calorie deficit of 1029 kcal and the largest drop in HDL, one can question if the kcal deficit should be kept at a lower level to avoid HDL falling below levels for primary prevention, and emphasize participation in exercise beyond weight loss, to increase HDL levels.

In conclusion, even with improvements in CVD risk factors in all intervention groups, a between-group comparison showed that exercise had beneficial effects on body composition in the NORM-EX group, in addition to causing the most favorable changes in total cholesterol and triglycerides. Nonetheless, the positive effect of exercise on CVD risk factors comparing the NORM-EX vs. NORM group was not detectable when comparing the LCHF-EX vs. LCHF group. The novelty of these findings is that the LCHF diet used in the study can blunt the aforementioned positive effect of exercise on CVD risk factors. However, even though total cholesterol and LDL were better in NORM-EX compared to LCHF-EX, both lipids reached levels below the primary prevention levels in the LCHF-EX group in response to the intervention. Therefore, the combination of LCHF diet and exercise used in this study is an adequate method to improve body composition and CVD risk factors in overweight and obese females.

## Figures and Tables

**Figure 1 nutrients-13-00110-f001:**
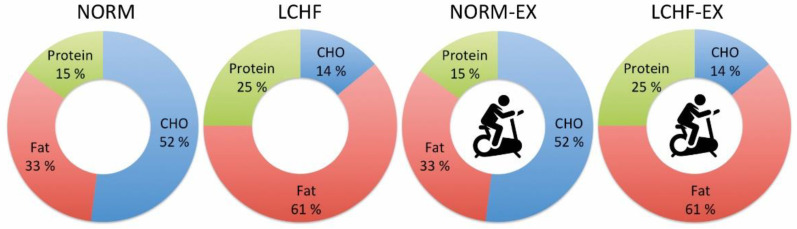
The E% for each macronutrient reflects the average ration planned for the 10 weeks. The carbohydrate content was increased gradually in the low-carbohydrate high-fat (LCHF) groups, with a proportional decrease in fat intake. Intervention groups. Diet only: Normal (NORM) and low-carbohydrate high-fat (LCHF) diet. Diet + exercise: Normal diet and exercise (NORM-EX) and low-carbohydrate high-fat diet and exercise (LCHF-EX). A weekly energy deficit of 4900 kcal was achieved either with calorie restriction or calorie restriction and exercise.

**Figure 2 nutrients-13-00110-f002:**
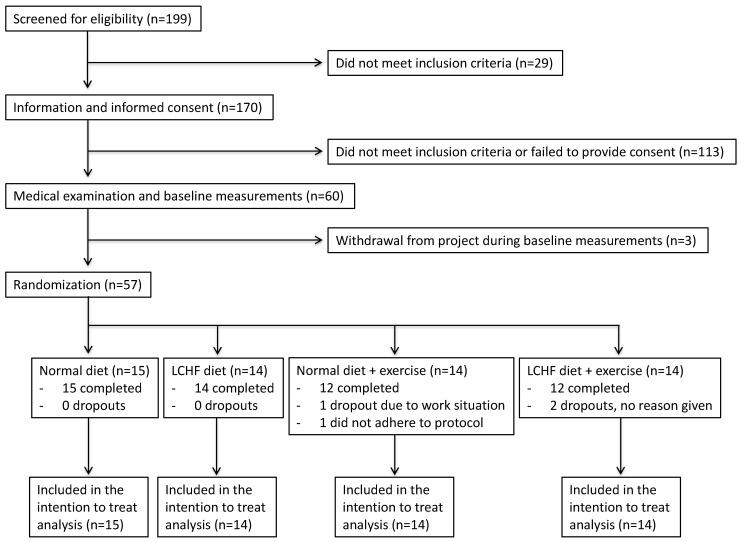
Flow diagram of enrollment and participant flow as recommended by CONSORT.

**Figure 3 nutrients-13-00110-f003:**
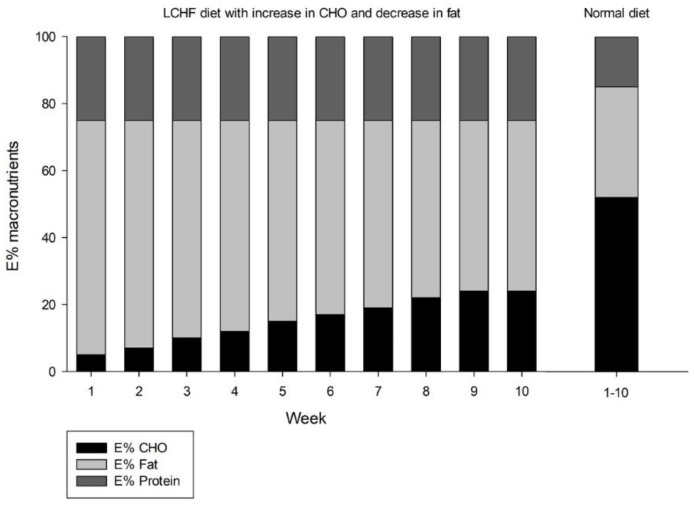
Target of E% of carbohydrates (CHO), fat and protein in the low-carbohydrate high-fat (LCHF) diet, and normal (NORM) diet (far right). In the LCHF groups, an increment of 10 g of carbohydrates was planned every week, until participants reached 100 g daily intake. A proportional decrease in kcal fat was planned alongside the carbohydrate increase, to keep kcal intake within protocol. The NORM diet was based on recommendations by the Norwegian Health Authorities and defined as normal diet for this study.

**Figure 4 nutrients-13-00110-f004:**
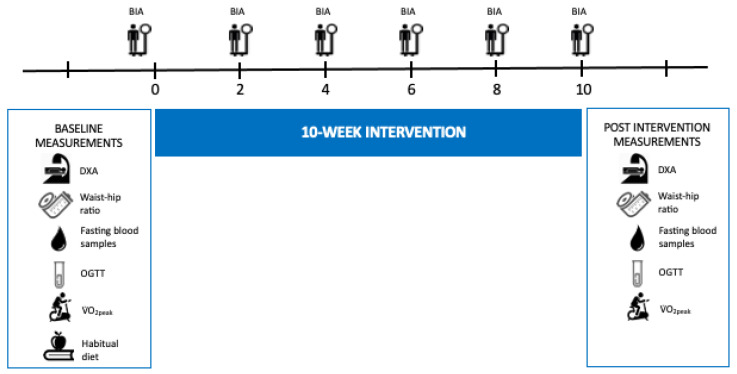
The figure shows measurements during baseline, intervention and post-intervention. During the baseline period, all participants registered their habitual diet for 4 days, and all groups kept dietary records throughout the intervention (not shown in the figure). A 2-week period has been highlighted to show exercise sessions for the exercise groups (NORM-EX and LCHF-EX) where the diet only groups (NORM and LCHF) continued their sedentary lifestyle. This procedure was performed throughout the intervention. All groups were weighed every second week using a BIA device. DXA: Dual-energy X-ray absorptiometry, OGTT: Oral Glucose Tolerance Test, VO_2peak_: Peak Oxygen Uptake, BIA: Bioelectrical Impedance Analysis.

**Table 1 nutrients-13-00110-t001:** Baseline characteristics.

	NORM (*n* = 15)	LCHF (*n* = 14)	NORM-EX (*n* = 14)	LCHF-EX (*n* = 14)
Age (years)	38.6 ± 3.7	40.0 ± 3.0	40.5 ±3.7	40.8 ± 3.3
Weight (kg)	89.2 ± 9.2	88.5 ± 7.2	89.4 ± 9.6	87.5 ± 11.2
Height (cm)	170.7 ± 5.2	169.2 ± 6.3	170.2 ± 4.5	166.4 ± 4.4
BMI (kg∙m^−2^)	30.7 ± 2.3	30.9 ± 2.6	30.8 ± 2.3	31.6 ± 3.0
Waist circumference (cm)	100.4 ± 6.0	99.1 ± 7.0	99.7 ± 5.2	101.6 ± 5.7
Hip circumference (cm)	114.3 ± 6.3	113.6 ± 6.5	114.8 ± 8.3	115.1 ± 8.1

Data are presented as mean ± standard deviation (SD). NORM: normal diet, LCHF: low-carbohydrate high-fat diet, NORM-EX: normal diet combined with exercise, LCHF-EX: low-carbohydrate high-fat diet combined with exercise, BMI: body mass index.

**Table 2 nutrients-13-00110-t002:** Body composition at baseline and after the 10-week intervention.

	NORM	LCHF	NORM-EX	LCHF-EX	NORM vs. LCHF	NORM vs. NORM-EX	LCHF vs. LCHF-EX	NORM-EX vs. LCHF-EX
Outcome Variables					Between Group Differences at Baseline vs. Post Intervention (*p*-Values) ^			
Body Composition	(n = 15)	(n = 14)	(n = 14)	(n = 14)				
Lean body mass (kg) pre	47.9 ± 4.4	48.1 ± 3.9	49.0 ± 4.5	45.9 ± 5.3	0.869	0.471	0.192	0.082
Lean body mass (kg) post	46.1 ± 4.3	47.2 ± 4.3	48.2 ± 5.0	44.2 ± 4.8	0.546	0.234	0.201	0.103
Within-group change (*p*-value)	**<0.001**	**<0.001**	**<0.001**	**0.001**				
Fat mass (kg) pre	38.9 ± 6.0	37.8 ± 5.7	37.5 ± 6.0	39.1 ± 6.5	0.588	0.551	0.550	0.519
Fat mass (kg) post	35.8 ± 6.1	31.5 ± 3.1	32.5 ± 7.2	33.5 ± 7.4	0.198	0.272	0.792	0.867
Within-group change (*p*-value)	**<0.001**	**<0.001**	**<0.001**	**<0.001**				
Fat % (tissue) pre	43.3 ± 5.0	42.5 ± 3.9	41.8 ± 3.5	44.5 ± 3.3	0.512	0.246	0.128	0.031
Fat % (tissue) post	43.5 ± 3.5	40.0 ± 3.0	40.0 ± 4.2	42.8 ± 4.6	0.063	**0.024 ***	0.477	0.315
Within-group change (*p*-value)	0.606	**<0.001**	**0.001**	**<0.001**				
Visceral fat (kg) pre	1.1 ± 0.5	0.9 ± 0.5	1.0 ± 0.5	1.2 ± 0.3	0.250	0.391	0.123	0.275
Visceral fat (kg) post	1.0 ± 0.5	0.6 ± 0.3	0.7 ± 0.3	0.9 ± 0.3	0.182	0.186	0.512	0.362
Within-group change (*p*-value)	**<0.001**	**<0.001**	**<0.001**	**<0.001**				
Waist-hip ratio (U) pre	0.88 ± 0.07	0.87 ± 0.07	0.87 ± 0.05	0.89 ± 0.06	0.793	0.650	0.607	0.412
Waist-hip ratio (U) post	0.86 ± 0.07	0.84 ± 0.07	0.84 ± 0.04	0.87 ± 0.04	0.391	0.339	0.302	0.180
Within-group change (*p*-value)	**0.044 ***	**0.002**	**0.002**	0.067				

Data are presented as mean ± standard deviation (SD). Linear mixed models, with pairwise comparisons were used to evaluate within- and between-group differences. Significant values are presented with bold numbers. ^ Differences between groups at post-measurements are adjusted for baseline measurements. * *p*-values marked with* are no longer significant with Bonferroni adjustment. NORM: normal diet, LCHF: low-carbohydrate high-fat diet, NORM-EX: normal diet combined with exercise, LCHF-EX: low-carbohydrate high-fat diet combined with exercise.

**Table 3 nutrients-13-00110-t003:** Cardiorespiratory fitness and lipid profile at baseline and after the 10-week intervention.

	NORM	LCHF	NORM-EX	LCHF-EX	NORM vs. LCHF	NORM vs. NORM-EX	LCHF vs. LCHF-EX	NORM-EX vs. LCHF-EX
Outcome Variables					Between Group Differences at Baseline vs. Post Intervention (*p*-Values) ^			
Cardiorespiratory Fitness	(n = 15)	(n = 14)	(n = 14)	(n = 14)				
$\dot{V}$O_2peak_ (mL∙min^−1^) pre	2497 ± 239	2490 ± 340	2478 ± 315	2259 ± 330	0.954	0.847	0.086	0.067
$\dot{V}$O_2peak_ (mL∙min^−1^) post	2364 ± 273	2356 ± 409	2715 ± 310	2416 ± 345	0.864	**0.001**	0.311	0.080
Within-group change (*p*-value)	**<0.01**	**<0.001**	**<0.001**	**<0.001**				
**Lipids**	**(n = 15)**	**(n = 14)**	**(n = 14)**	**(n = 14)**				
Total cholesterol (mmol∙L^−1^) pre	5.2 ± 0.7	5.1 ± 0.8	5.0 ± 0.8	5.6 ± 1.1	0.798	0.431	0.179	0.064
Total cholesterol (mmol∙L^−1^) post	4.7 ± 0.7	4.7 ± 1.0	3.9 ± 0.6	4.9 ± 1.1	0.958	**0.003**	0.513	**0.004**
Within-group change (*p*-value)	**<0.001**	**0.016**	**<0.001**	**<0.001**				
LDL (mmol∙L^−1^) pre	3.0 ± 0.6	3.1 ± 0.8	2.9 ± 0.8	3.5 ± 1.0	0.775	0.836	0.190	0.071
LDL (mmol∙L^−1^) post	2.7 ± 0.7	2.8 ± 0.8	2.3 ± 0.6	2.9 ± 0.9	0.698	0.066	0.544	**0.018 ***
Within-group change (*p*-value)	**0.005**	0.103	**<0.001**	**0.001**				
HDL (mmol∙L^−1^) pre	1.6 ± 0.3	1.5 ± 0.2	1.4 ± 0.4	1.5 ± 0.4	0.260	0.158	0.988	0.531
HDL (mmol∙L^−1^) post	1.3 ± 0.2	1.3 ± 0.2	1.1 ± 0.2	1.3 ± 0.3	0.527	**0.029 ***	0.576	0.177
Within-group change (*p*-value)	**<0.001**	**<0.001**	**<0.001**	**<0.001**				
TG (mmol∙L^−1^) pre	1.10 ± 0.48	1.06 ± 0.27	1.37 ± 0.76	1.45 ± 0.52	0.739	0.238	**0.004**	0.728
TG (mmol∙L^−1^) post	1.11 ± 0.34	1.00 ± 0.26	0.87 ± 0.21	1.03 ± 0.36	0.405	**0.030 ***	0.938	0.198
Within-group change (*p*-value)	0.952	0.583	**<0.001**	**<0.001**				
Total cholesterol/HDL ratio pre	3.3 ± 0.8	3.4 ± 0.8	3.7 ± 1.1	3.9 ± 1.0	0.663	0.202	0.207	0.748
Total cholesterol/HDL ratio post	3.6 ± 0.9	3.7 ± 1.0	3.6 ± 0.8	3.8 ± 0.8	0.640	0.980	0.962	0.534
Within-group change (*p*-value)	**0.007**	**0.004**	0.511	0.740				

Data are presented as mean ± standard deviation (SD). Linear mixed models, with pairwise comparisons were used to evaluate within- and between-group differences. Significant values are presented with bold numbers. ^ Differences between groups at post-measurements are adjusted for baseline measurements. * *p*-values marked with* are no longer significant with Bonferroni adjustment. NORM: normal diet, LCHF: low-carbohydrate high-fat diet, NORM-EX: normal diet combined with exercise, LCHF-EX: low-carbohydrate high-fat diet combined with exercise. $\dot{V}$O_2peak_: peak oxygen uptake, LDL: low-density lipoprotein, HDL: high-density lipoprotein, TG: triglyceride.

**Table 5 nutrients-13-00110-t005:** Attendance, average energy expenditure, intensity and rate of perceived exertion (RPE) during indoor bicycle exercise.

		NORM-EX				LCHF-EX				*p* Value		
	Expenditure Kcal	% HR_peak_	RPE	Attendance%	Expenditure Kcal	% HR_peak_	RPE	Attendance%	Expenditure Kcal	% HR_peak_	RPE	Attendance%
Exercise sessions *	(n = 12)	(n = 12)	(n = 12)		(n = 12)	(n = 12)	(n = 12)	%				
Week 1–2	524 ± 18	80 ± 3	15 ± 1	89 ± 15	513 ± 17	81 ± 3	15 ± 1	90 ± 1	0.119	0.338	0.738	0.811
Week 3–4	531 ± 22	79 ± 3	15 ± 1	79 ± 19	524 ± 24	80 ± 4	15 ± 1	89 ± 11	0.461	0.567	0.296	0.138
Week 5–6	537 ± 52	79 ± 4	16 ± 1	88 ± 19	531 ± 55	80 ± 3	16 ± 1	94 ± 11	0.791	0.653	0.602	0.283
Week 7–8	544 ± 35	78 ± 3	16 ± 1	88 ± 13	525 ± 60	79 ± 4	16 ± 1	90 ± 17	0.333	0.586	0.286	0.649
Week 9–10	551 ± 46	79 ± 4	17 ± 1	100 ± 0	530 ± 59	79 ± 4	17 ± 1	100 ± 0	0.351	0.796	0.432	1.000
Average	538 ± 29	79 ± 3	16 ± 1	88 ± 7	525 ± 40	80 ± 3	16 ± 1	93 ± 7	0.373	0.630	0.940	0.128

Data are presented as mean ± standard deviation (SD). Between-group differences were assessed using t-test with unequal variances. * Two-week intervals were used to calculate average for sessions. Participants exercised three sessions per week; NORM-EX: normal diet combined with exercise, LCHF-EX: low-carbohydrate high-fat diet combined with exercise, % HR_peak_: percentage of peak heart rate measured during testing of peak oxygen uptake, RPE: rate of perceived exertion on the Borg scale.

**Table 6 nutrients-13-00110-t006:** Average protein intake in grams per kilogram of body weight and gram per kilogram of Lean Body Mass (LBM) during the intervention.

	NORM	LCHF	NORM-EX	LCHF-EX	NORM vs. LCHF	NORM vs. NORM-EX	LCHF vs. LCHF-EX	NORM-EX vs. LCHF-EX
Protein Intake					Between Group Differences (*p*-Values)			
Protein g∙kg^−1^	0.9 ± 0.1	1.0 ± 0.1	0.9 ± 0.1	1.1 ± 0.1	**<0.001**	0.994	0.803	**<0.001**
Protein g∙kg LBM^−1^	1.6 ± 0.2	1.9 ± 0.2	1.5 ± 0.2	2.0 ± 0.2	**<0.001**	0.432	0.216	**<0.001**

Data are presented as mean ± standard deviation (SD). Between-group differences were assessed using t-test with unequal variances. Significant values are presented with bold numbers. NORM: normal diet, LCHF: low-carbohydrate high-fat diet, NORM-EX: normal diet combined with exercise, LCHF-EX: low-carbohydrate high-fat diet combined with exercise, LBM: lean body mass.

## Data Availability

The data that support the findings of this study are available on request from the corresponding author. The data are not publicly available due to privacy and ethical restrictions.

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
