# Peer review of "Low-Carbohydrate High-Fat Diet and Exercise: Effect of a 10-Week Intervention on Body Composition and CVD Risk Factors in Overweight and Obese Women—A Randomized Controlled Trial"

_nutrients, 2020, doi:10.3390/nu13010110_

Round 1

Reviewer 1 Report

The study entitled “Low-carbohydrate high-fat diet and exercise: Effect of 10-week intervention on body composition and CVD risk factors in overweight and obese women- A randomized controlled trials” by Valsdottir et al. describes a 10 wk 2x2 factorial intervention study where the effects of normal, LCHF, normal-exercise and LCHF-exercise hypocaloric regimes on weight, body composition and cardiometabolic risk factors were evaluated in overweight and obese women. The study addresses an interesting research topic- i.e. to what extent a hypocaloric regimen with different types of diets and/or exercise could have differential effects on the outcomes investigated. The study is well described and appears to have been well conducted. However it appears that the primary outcome has been adjusted after the study was finalized since there is discrepancies in the primary outcome defined (body composition) and what the study was actually powered for (glucose tolerarance!). This destroys the validity of the study and raises ethical considerations that must be addressed.

General comments and major concerns:

When looking up the clinicaltrials.gov record it seem that the primary outcome of the study was glucose tolerance, while the secondary outcome of the study was lipid profile. Body composition is not even stated as an outcome on clinicaltrilas.gov. Any changes in the outcomes, as well as justification for this, must be explained clearly in the manuscript. If some outcomes of the study have already been published elsewhere and the current manuscript is a secondary/exploratory analysis of additional outcomes, it needs to be explained as well. Furthermore, the inclusion criteria listed in clinicaltrials.gov does not match the criteria listed in the paper (age and BMI).

Sample size: Why do the authors base their sample size calculation on glucose tolerance when their primary outcome is changes in body composition? The authors don’t even measure glucose tolerance in this study! Furthermore, it is not clear which standard deviation authors assumed in their power calculation and if they took multiple comparisons into account when conducting the samples size calculation (if not, their study would have been underpowered).

When was the study conducted? In line 91-92 it says that the study was conducted January-April 2014, but in lines 272-273 it says that it was the follow-up that was conducted during this timeframe and that the recruitment (and baseline?) was conducted from October 2013. In clinicaltrials.gov it says that the first participant was enrolled in end of January 2021. If the authors started the recruitment in 2013 they need to be aware of the definition of “first participant enrolled” according to clinicaltrials.gov.

The authors present the rationale and importance of coming up with new strategies for weight-loss as means to improve public health through mitigation of NCDs. However, they do not acknowledge that weight-loss in the context of 10 wk is often transient and that most individuals will regain their weight within 0.5-1 year. What is desired is long-term weight loss and small weight-regain after a weight-loss. Please acknowledge this and put your results in this context.

Detailed comments:

Introduction:

Line 46: The reference [2] does not say anything about lean body mass being associated with obesity.

Line 63: The authors state that energy deficit needs to be 500-700 kcal/per day in order to lose weight, but any energy deficit will result in weight reduction. The energy deficit will determine the speed of the weight loss, but any energy deficit will induce a weight loss no matter how small or large it is. Please adjust.

Method:

It would have been nice with an overview (perhaps a table) of the study activities. It is quite hard to see from the text what is done and when.

Figure 2: The third box from above says “Screened: medical examination…” but from the text one gets the impression that this refers to baseline measurements? The authors need to clarify what is screening and what is baseline. Additional, from reading section 2.3 it is difficult to understand if the participants started the intervention immediately after the screening/baseline measurements or some time after the measurements (if so, how long after and what instructions did the participants receive during the “gap time”?).

Figure 3: Why did the target protein intake differ between the groups (10-20 E% vs 25 E%)? There has been many studies investigating the effect of protein on outcomes such as appetite and weight loss, so it could be speculated that a difference in protein intake could affect the results. Furthermore, it is difficult to understand the rationale for the increasing amount of carbohydrate in the LCHF group? The authors state that this is done to increase quality of life and compliance, but if the authors already knew beforehand that the very low carbohydrate would have a negative impact, why didn’t they plan a slightly higher (but stable) carbohydrate intake from the beginning? The authors need to elaborate on this strategy and explain why they use an increasing carbohydrate content rather than having the same carbohydrate content throughout the study. Additionally, it would be interesting to hear their thought on how this strategy would be used in a “real life” situation. Would the authors suggestion be that people eat LCHF diet for a pre-determined fixed period and therefore slowly increase the carbohydrate intake already from the beginning in order to read a “normal diet”.

Line 139: The authors need to explain the Borg scale and/or give some reference to it.

Line 177: The authors state that the weight was measured twice a week during the study period. Did the authors do this at the clinic or did the participants do it at home? Were the participants fasting, wearing clothes etc?

It would be interesting to see the bodyweights half way through the study.

Line 178: The authors state that the habitual intake was assessed at baseline prior to the study using a four-day food record. What did they assess? Was it a weighted food record?

Line 214: For how long were the samples stored at 4 C?

Line 228: What do the authros mean by “at a second time during the week”? Did they have 2 sessions per week?

Results:

Compliance: how was the compliance (diets) during the study period?

Table 1: According to the CONSORT guideline significance testing on baseline data should be avoided (http://www.consort-statement.org/checklists/view/32--consort-2010/510-baseline-data ).

Section 5.2: How do the authors define compliance? Is it enough to attend the session or did they have to reach the target energy expenditure? How did the authors handle participants who missed session? Did the authors just note their absence, called them a certain number of times or something else? The authors sate that data is not shown, but I would consider adding this data to table 6 (the authors could add a column called “attendance”).

Table 5: I think the row with “Expenditure” is a bit misleading. The calories available for the participants will be the calories they eat, regardless of what they use them for (e.g. exercise). Furthermore, “expenditure” means energy used, not energy available, so I think you need to reconsider this measure. If I understand correctly what you are trying to show I would call it “theoretical energy deficit” and calculate their theoretical energy deficit based their energy expenditure (including exercise) and their reported energy intake. Furthermore, the authors state that they collected dietary data every day during the intervention (line 220-221), but they only show “post-intervention” (in addition to the baseline intake). First of all, what does post-intervention mean in this context? Is it an average daily intake over the last intervention week, 4 weekdays and one weekend day in the end of the intervention period, or is it something else? Secondly, why don’t the authors show dietary data for more timepoints during the intervention? I think this is especially relevant since the authors instructed the participants to change the diet continuously throughout the intervention (increasing carbohydrate intake). Lastly, I would like to see the macronutrient in grams, since the authors partially instruct the participants based on this and it would be interesting to see how well they manage to reach the recommendation (e.g. grams of carbohydrate per day that goes from 20-100g over the 10 weeks).

Discussion:

Section 9. strengths and limitations: The authors mention that it could be a limitation that the non-exercise groups received less attention than the exercise groups. How were the non-exercise groups instructed to “behave” during the study; were they instructed to refrain from exercise, maintain their usual exercise habits or something else (and what were the exercise groups instructed to do, besides attending the bicycle classes)? Did the authors in any way monitor the participants exercise habits during the study, e.g. by repeating the baseline questionnaires you used to assess PAL? Is there a risk that the exercise groups may have reduced other habitual exercise, as they now attended the bike classes or maybe the non-exercise groups got inspired to be more active?

Limitations: The participants were instructed to keep dietary records for ten weeks which is major task for the participants. The authors briefly mention that self-reported dietary intake is often under-reported. Since the compliance is based on the dietary records I think that the authors should elaborate this much further.

Conclusion:

Be consistent in the terminology the authors use to describe their intervention groups; I assume they mean NORM when they write “diet” or “normal diet” in the conclusion?

The authors conclude that “In conclusion, exercise had beneficial effects on body composition in the normal diet group” but they did not find any differences between the groups in terms of body composition, so the main conclusion should be that there were no difference between the groups (but that they all induced an equal weight loss).

Reviewer 2 Report

The current manuscript assesses the effect of a 10 week low-carbohydrate diet and exercise on body composition and cardiovascular risk factors in obese women. This is a well-designed and well balanced trial however, the results of the study in its current form do not present any new or noteworthy findings on the field.

The authors should elaborate further on why they chose only females and why it would be important to explore how we could reduce cardiovascular risk in this population.

Highlighting what are the important and new findings of these study compared to the existing literature would also help significantly this manuscript.

Outlined below some suggestions that hopefully will improve the manuscript.

Abstract

The abstract is well articulated with a clear description of the design and outcomes.

27-30 please add SD in the presented data

Introduction

The introduction is concise but the not well justified. The exercise and diet variables are not connected in the intro well and there is no good justification of the literature gap. Authors should make clear through their introduction,

  • Why it is important to explore the effect of LCHF compared to normal diet?
  • Why it is important to explore the addition of the exercise factor?
  • What are the concerns of LCHF diet? i.e. cardiometabolic related concerns. This is not clearly outlined.
  • What will this study add to the literature? New and noteworthy. The literature gap is not clear.
  • Also the current study includes only female participants. Why? There is no justification of this in the intro. What is the physiological background of this decision? Are there any sex related differences in the effect of LCHF previously reported in the literature that justifies this decision?

72-80 The focus should be on improving cardiometabolic health. Losing weight is vague and can mean a lot of things not related to good health.

Methods

The current study is very well designed and the methods are well described. A very well balanced intervention

174- How did the authors control for the adherence of the participants?

236-239 Are the authors referring to beta-hydroxybutyrate concentrations? Please specify.

Results

The results are well presented.

Please add mean difference and SD in the data you present in text.

In the results sections you are often present post-hoc analyses whilst you have no between group interactions which is uncommon. Please clarify.

Results for energy expenditure and protein intake are reported both in text and in Tables 6 and 7. The results are already overtly long and these are not primary outcome data.  I suggest these data to be reported either in text or in tables.

Table 2 is redundant. Body weight is not a primary outcome. These information could be presented concisely in text.

Table 3. Please also report between group p values

Table 4. Please also report between group p values

Section 6.3- Are the authors referring to beta-hydroxybutyrate concentrations? Please specify.

How did the authors define ketosis levels?

Discussion

The authors present a good mechanistic discussion of their results. However, it is still not clear what is new and what this study added to the literature. A translational focus of the results would also help the current discussion.

442-448 The authors need to make clear that the focus of the study is clearly reflected in the text. The title, design and results of the current manuscript suggest that the authors aimed to assess the effect of the prescribed intervention on weight loss, CVD parameters and body comp. There is no correlation reported between weight loss and body comp or CVD parameters change. This needs to be rephrased.

Why a 5% loss is desirable? Please justify.

463-470 Please clarify if the results discussed are based on a relative (per kg) % change in LBM

482-490 Confusing structure. The authors are discussing regarding FM loss on the first paragraph of this section and then they revisit this topic in the last paragraph with contradictory arguments. Please clarify and be consistent.

594-599 This is not reported in the results section. Also this section lacks of discussion depth and references.

601-603 A good point that it is not discussed at all throughout the manuscript. There is a lack of discussion around the chosen population (females) and CVD risk.  This should be reflected throughout the manuscript.

Conclusion section should reflect the translational importance of the results. What this study adds to the literature? What did we learn that we didn’t know before?

Reviewer 3 Report

This manuscript reports the effects of weight-loss induced with a hypo-caloric “LCHF diet” (with a gradual increase in carbohydrates, and a proportional decrease in fat) or “normal diet” (a diet according to the Norwegian Health Authorities), with and without exercise, on body composition, cardiorespiratory fitness and CVD risk factors. The study is interesting and contributes with some new insights in the field. However, there are things in the manuscript that must be attended to.

Major concern: What is both novel, but at the same time problematic, in the study design is the gradual increase in carbohydrates (and a proportional decrease in fat) during the study's ten weeks. The LCHF diet week one is a very strict ketogenic LCHF diet, but the diet week ten is quite different with a daily intake of 100 g of carbohydrates (and thus >20% of total energy intake). This continuous change in the diet induce a continuous change in metabolic parameters, blood lipids, gut microbiota, etc. The question that arises is what in the diet itself causes the specific effects reported in the study? It is a challenge to discuss changes in blood lipids, for example, in relation to other diets when this particular LCHF dietary intervention is so different, with its continuously increasing carbohydrate content. It becomes very difficult to deduce specifically what it is in the diet that different outcomes relate to. These aspects must be discussed in detail in a revised version of the manuscript. In addition, the authors must make clear to the reader in all different parts of the manuscript (from abstract to discussion) that the so-called LCHF diet has a gradually increasing proportion of carbohydrates throughout the study´s 10 weeks.

Introduction:

Line 59-60: I think reference 13 is not that suitable, the two diets in that paper are low-calorie diets, but not really LCHF diets. It is thus a mismatch between reference 13 and what is written in line 59-60.

Line 79: I do not think it is correct to call the prescribed interval exercise program (performed as seven 4-minute intervals) in this study “endurance exercise”? Moreover, in the Methods section (line 85) it is called “high-intensity endurance exercise”. Try to be consistent throughout the manuscript.

Methods:

Line 86: Is it correct to term a diet according to the Norwegian Health Authorities as “normal” (NORM)? If possible, consider another name of the diet. Maybe “recommended” or “control” diet is better?

Figure 1: The figure legends lack too many details. For example, it has to be explained that the circles correspond to the average (?) macronutrient intake during 10 weeks (?) and that there was a gradual increase in carbohydrates during the study's ten weeks (for the LCHF diet). Is it the actual intake, or planned intake? The abbreviations (NORM, NORM-EX, etc) must be explained. Finally, the last sentence should be deleted, it does not really fit here.

Section 2.1, Line 102: Is the BMI as well as the age interval correct (it differs from what is written in ClinicalTrails NCT04100356)?

Line 103-104: Has thyroid status been examined? Is it not un-common with this type of endocrine disorder in this kind of study population (middle-aged women).

Figure 2: How come there were subjects who did not meet inclusion criteria AFTER the researcher received informed consent? This seems strange.

Section 2.3, Line 124: Please clarify that -3400 kcal/week depends on the diet and that -1500 kcal/week depends on the exercise.

Section 2.5 and 2.6: Did the participants get any recipes? If not, how could, for example, individual participants manage to target a 70% fat intake in the first week of the LCHF intervention, without being experts in nutrition?

Figure 3: Explain in figure legend what a “Normal diet” is.

Section 2.7, Line 176-177: What is the time-point for the post-intervention data collection?

Line 178: Do you have some data on their habitual dietary intake? It would be interesting to see if their normal habitual dietary intake differed from the so-called NORM diet in the study.

Section 3.2: What was the time-point for DXA measurement in this study? Time of the day? Fed or fasted? Same for all participants?

Section 3.3: What was the time-point for peak oxygen uptake measurement in this study? Time of the day? Fed or fasted? Same for all participants?

Section 3.5, Line 214: For how long were the samples stored at 4 C?

Section 3.6, Line 233: Could you specify how many nutritionists (in total) were involved in the study?

Section 4. I definitely think it should be stated somewhere in the manuscript what was the primary outcome of this trial. This must be clear to the reader of the manuscript. For example, the authors could simply add ”…, since area under curve (AUC) glucose was the primary outcome measure of the intervention” in line 243. Or a similar statement somewhere in the manuscript.

Results:

Table 1-7: Abbreviations must be explained and statistical tests are missing in most tables. See instructions from the journal.

Section 5.3.2, Line 313-314: There was a difference, but what was the difference (increase/decrease)?

Table 2: kg is missing.

Section 6.3: The authors described that ketone bodies were monitored every morning using urine sticks (Methods section, line 235-239). How did the authors come to the reported urine ketone levels in line 420-421 using semi-quantitative urine sticks? Moreover, was the presence of ketosis measured in the groups with normal diet (with or without exercise)? This could be interesting, since all groups were in a catabolic state.

Table 7: The results in Table 7 should be referred to in the Results section. At the moment, these results are referred to only in the Discussion section.

Discussion:

Line 442: According to the ClinicalTrials protocol the primary outcome of the whole trial was area under curve (AUC) glucose. The statement “The present study was designed to assess the effect of… on body composition and CVD risk factors…” in this regard is actually a bit misleading. Consider rewriting.

Line 449: According to the ClinicalTrials protocol the primary outcome of the whole trial was area under curve (AUC) glucose. Both body composition and CVD risk factors reported in this paper are secondary outcomes.

Minor:

Line 127: Skip “_”.

Line 205: Remove an extra space.

Line 631: Replace “An” with “A”.

Reviewer 4 Report

This is good trial.

1) Line 261, you treated missing value mean imputation.
This treating possibly including biases. Multiple imputation is recommended.

2) In Fig.2 you mentioned "intention to treat", but no mentioned in your methods and results. Please explain.  

3)your results showed decreasing HDL, too.
Originally, HDL should be able to increase by endurance exercise,  which you mentioned in  544– lines. Along with the results, it looks like no change of LDL/ HDL ratio which is stronger as a CVD risk factors.

Round 2

Reviewer 1 Report

The authors have addressed most of the issues I raised in an excellent way and the paper has been very much improved after the review. However, I still have problems with the way the endpoints are handled in relation to the previous main study. 

I suggest that authors makes it crystal clear in their aim already that the aim is to investigate the effects on SECDONARY endpoints of their original study. It is very confusing when the authors now refer to weight change as the PRIMARY outcomes in the conclusions for example. The results of the primary outcome (glucose tolerance) has already been published. It is totally wrong and misleading to "adjust" the outcome/endpoint hierarchy from that the study was designed for. 

I also suggest that authors make a posthoc power calculation and show the readers the actual power under the given conditions. Always good to know, since the study was not designed primerly for the outcome addressed here.

Author Response

Thank you for an excellent review of our paper and we appreciate the time an effort that you have put into giving us valuable feedback. We believe that the manuscript has improved from the adjustments, and hope you now find it suitable for publication in Nutrients.

Comment 1: I suggest that authors makes it crystal clear in their aim already that the aim is to investigate the effects on SECDONARY endpoints of their original study. It is very confusing when the authors now refer to weight change as the PRIMARY outcomes in the conclusions for example. The results of the primary outcome (glucose tolerance) has already been published. It is totally wrong and misleading to "adjust" the outcome/endpoint hierarchy from that the study was designed for. 

Answer 1: We agree that we have not used primary and secondary aims and outcomes in a correct way. This has been confusing for the reader although our intention was not to mislead the reader or to adjust the endpoint. This was solely a mistake. As you have requested, we have clarified in the introduction that this paper reports results of secondary endpoints from the main study (line 97-99). In addition, we have made it clear that the sample size calculation was made on AUC glucose, the primary endpoint in the study (line 304-313). We have used track changes to highlight where we have deleted “primary” and “secondary” outcome (line 332, 360, 391, 488, 534, 687, 694, table caption 2 and table caption 3).

Comment 2: I also suggest that authors make a posthoc power calculation and show the readers the actual power under the given conditions. Always good to know, since the study was not designed primarily for the outcome addressed here.

Answer 2: We have used a very conservative statistical analysis and have been clear in the limitations section that the sample size is small. We aimed for clinically relevant effect size in AUC glucose, as we knew that smaller changes would demand larger sample size. We agree that it is good to know the power under given conditions, and an option could be to perform a post hoc calculation to evaluate the power in the current paper. However, according to several statistical papers and discussions on the topic, there is little merit in a post hoc calculation of statistical power using observed effects of a trial because the significance of a test also determines the observed power (details in Hoenig & Heisey). We report p-values, which are directly related to the observed power. Results with statistical non-significant p-values will always show that a result will appear to have low observed power. For more details and figures see  https://linkinghub.elsevier.com/retrieve/pii/S0022480420305023. In line with this, we believe calculating observed power can be misleading and will not augment the information already given.

Please see the following literature for further details:
* Goodman SN, Berlin JA. The use of predicted confidence intervals when planning experiments and the misuse of power when interpreting results. Ann Intern Med. 1994;121: 200e206. 
* Hoenig JM, Heisey DM. The abuse of power: the pervasive fallacy of power calculations for data analysis. Am Stat. 2001;55:19e24.
* Lenth R. Some practical guidelines for effective sample size determination. Am Stat. 2001;55:187e193.
* Levine M, Ensom MH. Post hoc power analysis: an idea whose time has passed? Pharmacotherapy. 2001;21:405e409.
* O'Keefe DJ. Post hoc power, observed power, A priori power, retrospective power, prospective power, achieved power: sorting out appropriate uses of statistical power analyses. Commun Methods Meas. 2007;1:291e299.
* Lakens D. Observed power, and what to do if your editor asks for post-hoc power analyses. 2014.

* https://dirnagl.com/2014/07/14/why-post-hoc-power-calculation-does-not-help/

* http://daniellakens.blogspot.com/2014/12/observed-power-and-what-to-do-if-your.html

Our warmest wishes for a happy holiday season.

On the behalf of the authors

Ditta Valsdottir

Reviewer 2 Report

I am happy with all the changes the authors made. The paper has been significantly improved with the discussion reflecting well its results.

Author Response

Thank you for an excellent review of our paper and we appreciate the time an effort that you have put into giving us valuable feedback. We believe that the manuscript has improved from the adjustments, and hope you now find it suitable for publication in Nutrients.

Our warmest wishes for a happy holiday season.

On the behalf of the authors

Ditta Valsdottir